# The Effectiveness of Molecular, Karyotype and Morphological Methods in the Identification of Morphologically Conservative Sibling Species: An Integrative Taxonomic Case of the *Crocidura attenuata* Species Complex in Mainland China

**DOI:** 10.3390/ani13040643

**Published:** 2023-02-12

**Authors:** Haotian Li, Yaoyao Li, Masaharu Motokawa, Yi Wu, Masashi Harada, Yuchun Li

**Affiliations:** 1College of Life Sciences, Liaocheng University, Liaocheng 252059, China; 2Marine College, Shandong University (Weihai), Weihai 264209, China; 3College of Life Sciences, Linyi University, Linyi 276005, China; 4The Kyoto University Museum, Kyoto University, Kyoto 606-8501, Japan; 5School of Life Sciences, Guangzhou University, Guangzhou 510006, China; 6Laboratory Animal Center, Osaka City University, Osaka 545-8585, Japan

**Keywords:** *Crocidura attenuata* species complex, geographical distribution, integrated taxonomy, karyotype, molecular, morphology

## Abstract

**Simple Summary:**

Morphological conservation has always been a problem in the identification of sibling species. In this study, molecular, karyotype and morphological methods are integrated to revise the species classification and distribution of the *Crocidura attenuata* species complex in mainland China. The results show that there are five species in the *C. attenuata* species complex. In addition to four known species, namely, *C. attenuata*, *C. tanakae*, *C. anhuiensis* and *C. dongyangjiangensis*, there was an undescribed species distributed in Guangxi. Among them, *C. tanakae* is widely distributed in Southern China, *C. attenuata* is situated only around the Sichuan Basin, and *C. anhuiensis* and *C. dongyangjiangensis* are present in Southeast China. The effectiveness of molecular, karyotypic and morphological methods in the taxonomy of the *C. attenuata* species complex is also discussed in the present study, which will provide a reference for the taxonomic study of other morphologically conserved species.

**Abstract:**

The conservation of morphology has resulted in considerable issues in the taxonomy of small mammals, especially for the identification of sibling species. Moreover, it is often difficult to completely solve such taxonomic problems by relying only on a single research method. The genus *Crocidura* is one of the genera with a conservative morphology and high species diversity. Among them, *Crocidura attenuata* has been considered in the field as the most widely distributed and common species. In fact, it is a species complex containing multiple species, and the classification and distribution of this species is controversial. In this study, the species and distribution of the *Crocidura attenuata* species complex experienced an integrated revision using three different levels of research methods: molecular, karyotype and morphology. The results show that (1) the *C. attenuata* species complex contains four known species (*C. attenuata*, *C. tanakae*, *C. anhuiensis* and *C. dongyangjiangensis*) and a cryptic species distributed in Guangxi, which may be the same undescribed species as the “*C. attenuata*” distributed in Vietnam. (2) *C. attenuata* is only distributed around the Sichuan Basin, *C. tanakae* is the most widely distributed throughout Southern China, and *C. anhuiensis* and *C. dongyangjiangensis* are almost sympatric in Southeast China. Furthermore, (3) although the molecular method lacks a unified threshold for species classification, it can rapidly and effectively identify the species of the *C. attenuata* species complex. Although karyotype and morphology methods cannot completely solve the species classification issues in respect of the *C. attenuata* species complex, they can provide supplemental information for taxonomic purposes. Therefore, the integrated taxonomic method can present the advantages of different methodological levels, and will provide further evidence for the taxonomy of sibling species with a conservative morphology.

## 1. Introduction

One of the difficulties of taxonomy is the conservation of morphology among sibling species. However, the use of morphological methods alone results in the underestimation of the species diversity of morphologically conservative groups, especially in small mammals, which leads to species complexes, including cryptic and sibling species [1,2,3,4,5,6]. Therefore, integrated taxonomy based on molecular, cellular and morphological methods has been increasingly applied in the literature to perform the taxonomy of species complexes [7,8,9,10].

The genus of white-toothed shrews, *Crocidura*, is the most speciose genus of mammals, with 197 recognized species to date [11]. According to the statistics of the mammal diversity database of the American Society of Mammalogists (URL: https://www.mammaldiversity.org/, accessed on 31 December 2022), the genus *Crocidura* increased to 203 species in 2022. The taxonomy and distribution of *Crocidura* are complex and inconclusive because of their high diversity level and generally conservative morphology [12,13,14]. In China, the latest catalogue of mammals includes 14 species of *Crocidura*, comprising of *C. anhuiensis*, *C. attenuata*, *C. dongyangjiangensis*, *C. tanakae*, *C. lasiura*, *C. shantungensis*, *C. wuchihensis*, *C. vorax*, *C. suaveolens*, *C. rapax*, *C. dracula*, *C. sibirica*, *C. indochinensis* and *C. tadae* [15]. Although this catalogue contains most species of *Crocidura*, according to the multilocus molecular result, the species diversity of *Crocidura* in China remains underestimated in the literature [16].

*C. attenuata* has always been regarded as the most common species of *Crocidura* in China and is widely distributed in Southern China and surrounding countries and regions [17]. Due to the high conservation of its morphology, *C. attenuata* has become a species complex that includes both cryptic and sibling species. One of the most important cryptic species is *C. tanakae*, which has long been regarded as a Taiwanese subspecies or synonym of *C. attenuata* due to their extremely similar morphology [18,19,20,21,22,23]. In 2001, *C. tanakae* was again promoted to the status of an effective species and was only confirmed as distributed in Taiwan based on the chromosome data [24]. Subsequently, the status of *C. tanakae* as a distinct species was also supported by the molecular data [25]. In recent years, with the extensive application of molecular technology, it has been observed that *C. tanakae* is widely distributed in Vietnam, Laos and Philippines, as well as Hunan and Guizhou in Southern China [26,27,28,29]. Li et al. [30] revised the distribution of *C. attenuata* and *C. tanakae* in mainland China based on molecular methods, and observed that the distribution of *C. attenuata* is apparently limited to around the Sichuan Basin and Southeast China. The natural range of this species is much smaller than that of *C. tanakae,* which is distributed almost all over Southern China. Therefore, many specimens of *C. attenuata* that have been reported to be widely distributed in Southern China and Southeast Asia are in fact *C. tanakae*. Recently, two new species of *Crocidura* were reported in the Anhui and Zhejiang Provinces of China, namely *C. anhuiensis* [31] and *C. dongyangjiangensis* [32]. They are distributed in areas that have experienced relatively sufficient investigations and are sympatrically distributed with *C. attenuata*. However, the main reason they have not been described until recently may be that they have been regarded in the literature as the cryptic species of *C. attenuata*. *C. anhuiensis* and *C. attenuata* are sibling species at the molecular level and are very similar in morphology [31]. Although the body size of *C. dongyangjiangensis* is smaller than that of *C. attenuata*, it is also similar to *C. attenuata* in other morphological characteristics, and the relatively low numbers of *C. dongyangjiangensis* may have been regarded in the literature as the juveniles of *C. attenuata.* Therefore, we believe that *C. attenuata* populations widely distributed in Southern China are actually a species complex containing at least *C. attenuata*, *C. tanakae*, *C. anhuiensis* and *C. dongyangjiangensis.* It is also necessary to revise the ambivalence regarding the species classification and distribution of the *C. attenuata* species complex.

In this study, molecular, karyotype and morphological methods were integrated to conduct taxonomic research on the *C. attenuata* species complex to revise its species classification and distribution. Moreover, the effects of different research methods on the taxonomy of related species with a conservative morphology were discussed.

## 2. Materials and Methods

The specimens of the *Crocidura attenuata* species complex used in this study were obtained using a combination of Sherman live cages and pitfall traps during small-mammal surveys conducted in mainland China from the years 2000 to 2020 (Figure 1). A total of 242 specimens were used in this study, including 53 from *C. attenuata*, 164 from *C. tanakae*, 12 from *C. anhuiensis* and 13 from *C. dongyangjiangensis*. All specimens were deposited with the Zoological Research Team at Marine College, Shandong University (Weihai), China, and most of the specimens consisted of a carcass (stored in 95% alcohol), dried skin and a cleaned skull. Detailed information on the specimens is listed in Appendix A. The field methods followed the relevant Chinese laws, and all animals were handled in a manner consistent with the guidelines approved by the American Society of Mammalogists [33].

### 2.1. Molecular Analysis

We amplified and sequenced the complete mitochondrial protein-coding gene cytochrome *b* (*Cytb,* 1140 bp) obtained from 65 individuals (19 from *C. attenuata*, 28 from *C. tanakae*, 5 from *C. anhuiensis* and 13 from *C. dongyangjiangensis*) following the protocol described by Li et al. [30] and using a mammalian universal primer modified by our laboratory, including M13: GTAAAACGACGGCCAGTCCAATGACATGAAAAATCATCGTT and M14: CAGGAAACAGCTATGACTCTCCATTTCTGGTTTACAAGAC. Simultaneously, we used 177 *Cytb* sequences that had been previously sequenced in our laboratories [30,34]. To revise the species classification and distribution of the *C. attenuata* species complex, we downloaded 101 available *Cytb* sequences of the *C. attenuata* species complex (20 for *C. attenuata*, 57 for *C. tanakae*, 6 for *C. anhuiensis* and 18 for *C. dongyangjiangensis*) distributed in China for our phylogenetic analyses (Appendix A).

The sequences were manually edited in BioEdit v.7.2.5 [35] and aligned and examined using MEGA X software [36]. The best model of sequence evolution estimated by the Bayesian information criterion (BIC) in jModelTest v.2.1.4. [37] was GTR + G + I. We created the phylogenetic trees using the maximum likelihood (ML) in MEGA X software [36] and Bayesian inference (BI) in MrBayes 3.2 [38] based on the sequences of all *Cytb* haplotypes, with *C. Shantungensis* and *Suncus murinus* as the outgroups. The bootstraps were obtained using a rapid bootstrapping algorithm with 1000 replicates. For the Bayesian analysis, we performed four Markov chain Monte Carlo (MCMC) runs with 4 chains for 10,000,000 generations, sampling every 1000 trees and discarding the first 25% as burn-in. The interspecific and intraspecific genetic distances were calculated using the Kimura 2-parameter model implemented in MEGA X software [36].

### 2.2. Karyotype Analysis

According to the latest taxonomic catalogue of the *Crocidura* genus in China, the karyotypes of *C. attenuata*, *C. tanakae* and *C. anhuiensis* were reported in our previous studies [34]. Therefore, we only needed to analyze the karyotypes of *C. dongyangjiangensis*. Preparation of conventional karyotypes was conducted using femoral bone marrow in the field [39,40,41]. CytoVision System (Applied Imaging, Newcastle upon Tyne, UK) Software was used to shoot and analyze chromosomes with good dispersion outcomes in the mitotic phase. Each specimen was observed and counted for 30–80 metaphase cells, and the mode was used to determine the diploid chromosome number (2 n) and fundamental chromosome number (FN). The karyotypes of 5 *C. dongyangjiangensis* specimens were analyzed, including 3 obtained from Guangdong Province (G09247, G09248 and G12189) and 2 from Zhejiang Province (S1413 and S1431). (Appendix A).

### 2.3. Morphological Analysis

External measurements were taken directly in the field. Four measurements with a small coefficient of variation (CV < 10%) were selected: head and body length (HB), tail length (Tail), ear length (Ear), and hind-foot length with claw (HF). We identified 65 adults (16 from *C. attenuata*, 30 from *C. tanakae*, 8 from *C. anhuiensis* and 11 from *C. dongyangjiangensis*) for the morphological analysis and excluded individuals with severe skull damage and juveniles. Juveniles were excluded according to the degree of tooth wear and the presence of fused basioccipital and basisphenoid bones [42,43,44,45].

All 20 cranial characters were measured using an electronic digital caliper graduated to 0.01 mm. Among them, the following 13 measurements were identical to those presented by Hutterer et al. [45]: condylo-incisive length (CIL), height of cranial capsule (HCC), rostrum width (RW), maxillary breadth (MB), least interorbital width (IO), greatest width of skull (GW), upper toothrow length (UTR), length of anterior tip of P4 to posterior border of M3 (P4–M3), breadth of palate between the buccal margins of second molars (PW1), postglenoid width (PGL), length of lower molar series (m1–m3), length of mandible from tip of incisor to posterior edge of condyle (ML) and height of coronoid process (COR). Five measurements were identical to those from Meegaskumbura et al. [46]: length of maxillary tooth row (MTR), palatilar length (PAL), postpalatal length (PPL), length of dentary teeth excluding incisors (LDT1), and length of dentary teeth including incisors (LDT2). The remaining two measurements were the same as those presented in the study conducted by Jiang and Hoffmann [12], those being palato-incisor length (PIL) and breadth of coronoid process (BCP).

We calculated the mean and standard deviation of all external and skull morphological measurements, tested a univariate analysis of variance (ANOVA) and multiple comparisons, and conducted principal component analyses (PCAs) using SPSS Statistics 24.0 (SPSS, Chicago, IL, USA). Due to the likely existence of considerable interobserver variations in the external measurements, 3 PCA analyses were conducted based on 4 external, 20 cranial and all 24 morphological measurements.

## 3. Results

### 3.1. Sequence, Phylogeny and Genetic Distances

In this study, we successfully amplified the complete 1140-bp mitochondrial *Cytb* sequence for 65 individuals, with GenBank numbers OP594738-OP594802. Based on the *Cytb* data previously reported, we obtained 343 complete *Cytb* sequences for *C. attenuata* species complexes, which were divided into 132 haplotypes (Appendix A).

The BI and ML molecular phylogenetic trees presented nearly identical results. Although the phylogenetic tree did not describe the interspecies relationships of *C. attenuata* species complexes well, it had considerable support at almost the species level (BS = 100, PP = 1.00), with the exception of *C. anhuiensis*, which ensured the accuracy of the species identification of these specimens (Figure 2). According to the phylogenetic position of the topotype for *C. attenuata*, and the holotype of *C. anhuiensis* and *C. dongyangjiangensis*, we correctly classified the species for the specimens of the *C. attenuata* species complex and observed that more than half of the specimens belonged to *C. tanakae*. The four reported species all formed a monophyletic lineage, among which the closest relative to *C. attenuata* was *C. anhuiensis*, followed by *C. dongyangjiangensis*. Significantly, we observed that the samples of *C. attenuata* distributed in Southeast China in some previous reports were clustered in the same lineage as *C. anhuiensis* in the phylogenetic tree (Appendix A), which also verified that *C. anhuiensis* has always been a cryptic species of *C. attenuata.* In addition, the sequence FJ814039 of “*C. attenuata*” distributed in Guangxi gathered into a single lineage, forming a sister relationship with *C. attenuata.*

The interspecific genetic distance of the four species was greater than 10%, except for *C. attenuata* and *C. anhuiensis*, which were smaller (4.1%), while the intraspecific genetic distance of the four species was approximately 1% (Table 1). Therefore, the molecular method can rapidly identify the four species of the *C. attenuata* species complex.

### 3.2. Karyotype and Karyogram

The conventional karyotype of 5 individuals of *C. dongyangjiangensis* had a uniform arrangement of diploid chromosome number (2 n) = 40 and fundamental chromosome number (FN) = 54, 6(m + sm) + 6 st + 26 t + X(sm) + Y(t), with 3 pairs of metacentric or submetacentric, 3 pairs of subtelocentric, and 13 pairs of acrocentric autosomes, a submetacentric X chromosome, and an acrocentric Y chromosome (Figure 3).

Li et al. [34] reported that *C. tanakae* has karyotype polymorphisms and observed 32 karyotypes (2 n = 24−40, FN = 45−56) in 82 individuals, and the karyotypes of 11 *C. attenuata* were consistent, which means 2 n = 40, FN = 54. According to our molecular results, 5 of the 11 *C. attenuata* should belong to *C. anhuiensis*. Therefore, the karyotype of *C. dongyangjiangensis* is the same as that of *C. attenuata* and *C. anhuiensis*, which is consistent with the common karyotype of the genus *Crocidura* in China (Appendix A). Therefore, it is almost impossible to accurately classify the species of the *C. attenuata* species complex based on the karyotype alone, but some specimens of *C. tanakae* can be identified by some of its unique karyotypes.

### 3.3. Metrological Morphology and Morphological Characteristics

The external and skull measurement indices of 65 adult *C. attenuata* species complex specimens were obtained (Table 2), and almost all measurement indices conformed to a normal distribution (*p* > 0.05). Therefore, we conducted a univariate analysis of variance (ANOVA) and multiple comparisons on four species of the *C. attenuata* species complex using parameter statistical analysis. The ANOVA results show that almost all indices, except ear length, present significant morphological differences among the species (Appendix A). The results of multiple comparisons show that the measurement indices of *C. dongyangjiangensis* are significantly smaller than those of the other three species in *C. attenuata* species complex, belonging to the smaller shrew (Appendix A). *C. attenuata*, *C. tanakae* and *C. anhuiensis* are similar in most indices belonging to the medium-sized shrew (Table 2 and Appendix A). Overall, among the three species, *C. anhuiensis* is slightly larger, followed by *C. attenuata*, and *C. tanakae* is the smallest. The main performance is that *C. anhuiensis* is significantly larger than *C. tanakae* in all indices, except head and body lengths, and is significantly larger than *C. attenuata* in 15 measurement indices. *C. attenuata* is more similar to *C. anhuiensis* in its external morphology, which shows that none of their external indices are significantly different. However, it is closer to *C. tanakae* in terms of the skull indices because there are 9 skull indices that have no significant differences from *C. tanakae* and only 4 skull indices that have no differences from *C. anhuiensis* (Appendix A).

Although the statistical analysis determined that there were significant interspecific differences in most of the measurement indices of the *C. attenuata* species complex, there were many overlaps between species in the range of all the measurement indices. Therefore, according to the numerical value of the measurement indices, it is difficult to identify the species of the *C. attenuata* species complex, especially *C. attenuata*, *C. tanakae* and *C. anhuiensis*.

In the three PCA analyses, the first two principal components accounted for 76.37%, 88.48% and 80.91% of the variation based on 4 external, 20 cranial and all 24 morphological measurements, respectively (Appendix A). Almost all the variables were strongly correlated with the first principal component, which was mainly related to the size factor, and the skull indices presented a stronger correlation than the external indices. From the main scatter plots of the PCA, it can be observed that the skull indices are better at distinguishing the four species of the *C. attenuata* species complex, while the external indices are the worst (Figure 4). The external main scatter plots can only distinguish *C. dongyangjiangensis* from *C. attenuata* and *C. anhuiensis*. At the same time, it was observed that the external indices of *C. tanakae* were highly variable and overlapped with those of the other three species. The results of the skull main scatter plots can completely distinguish *C. dongyangjiangensis* and can also distinguish most samples of the other three species. The distinguishable effect of the combined indices of skull and external measurement is similar to that of the skull, but the effect is slightly worse. In conclusion, the PCA of the *C. attenuata* species complex can distinguish the smaller *C. dongyangjiangensis*, but cannot completely distinguish the other three species.

In terms of the external morphological features, the four species of the *C. attenuata* species complex are extremely similar, except for *C. dongyangjiangensis*, and it is difficult to observe significant differences between species (Appendix A). Compared with the other three species, *C. dongyangjiangensis* is significantly smaller, darker in pelage color and thinner in the tail with shorter hair. The most obvious feature of *C. dongyangjiangensis* is that the palms and soles of the feet usually feature many small, black granules, while the fore and hind feet of *C. attenuata*, *C. tanakae* and *C. anhuiensis* have little pigmentation, and the palms and soles of the feet are relatively smooth without obvious granular protrusions (Appendix A). Although the somatotype of *C. tanakae* is similar to that of *C. attenuata* and *C. anhuiensis*, there are only differences in its tail and hind-foot lengths (Table 2): the tail (tail/head–body length = 69 ± 8%, *n* = 28) and hind-foot (12.97 ± 0.65 mm, *n* = 30) lengths are relatively short, while the tail and hind-foot lengths of *C. attenuata* and *C. anhuiensis* are 76 ± 8% (*n* = 16) and 77 ± 8% (*n* = 8), 14.19 ± 1.09 mm (*n* = 16) and 14.76 ± 0.70 mm (*n* = 8), respectively. However, no obvious morphological characteristics were observed in the external characteristics of *C. attenuata* and *C. anhuiensis*.

Regarding the characteristics of the skull of *C. dongyangjiangensis*, it is also significantly smaller, while the sizes of *C. attenuata*, *C. tanakae* and *C. anhuiensis* are similar (Figure 5). In addition, the most upper dentitions of *C. dongyangjiangensis* is closely spaced (7/9 = 77.8%), especially P4 and U3, so that the anterior margin of P4 covers almost the entire posterior margin of U3, which is closer than *C. attenuata*, *C. tanakae* and *C. anhuiensis*. However, in most specimens of *C. attenuata* (15/16 = 93.8%), *C. tanakae* (26/30 = 86.7%) and *C. anhuiensis* (7/8 = 87.5%), there is a small gap between P4 and U3, or the two just touch. Moreover, the anterolingual margin of P4 in *C. dongyangjiangensis* is closest to the midline of the skull, and the protocone is almost parallel to the inside of the paracone, while in *C. attenuata*, *C. tanakae* and *C. anhuiensis*, the closest position to the central axis of the skull is the middle lingual margin of P4, and the protocone is in the posterior position of the paracone. The basioccipital region of most *C. tanakae* specimens is relatively flat and broad, while most of the *C. attenuata*, *C. anhuiensis* and *C. dongyangjiangensis* specimens are similar, and are narrow and ridged. The skulls of *C. anhuiensis* are stronger than those of the other three species in both the maxilla and mandible, and are significantly larger than those of *C. attenuata* and *C. tanakae* in many skull measurement indices (Appendix A). Moreover, the suborbital foramina of *C. anhuiensis* are slightly larger than those of the other three species. However, most of the interspecific morphological differences are based on the majority of the samples, rather than the absolute differences between species.

In conclusion, morphological methods can intuitively and effectively identify *C. dongyangjiangensis* in the *C. attenuata* species complex, but it is difficult to accurately identify the other three species.

## 4. Discussion

### 4.1. Species Classification and Distribution Revision of the C. attenuata Species Complex

In this study, we mainly integrative revised the four species discovered in the *C. attenuata* species complex using molecular, karyotype and morphological methods. There is no obvious difference in morphology between *C. attenuata* and *C. anhuiensis*, the karyotype is also consistent, and the genetic distance between them is less than that between other species (Appendix A and Table 1). However, they are clustered into separate clades, and their interspecific genetic distance (4.1%) is also close to the empirical threshold mammalian *Cytb* gene interspecific gap of 5% [47,48]. Compared with the smaller intraspecific genetic distance (Table 1), this indicates that there are considerable genetic differences between them, thus supporting *C. anhuiensis* as a distinct species.

It was also determined in the molecular phylogenetic tree that “*C. attenuata*” (FJ814039) distributed in Guangxi gathered into a separate clade and formed a sister lineage with *C. attenuata* (Figure 2). The BLAST results show that the similarity between the sequence and *C. attenuata* is less than 96%, while the sequence is the most similar (98%) to “*C. attenuata*” distributed in Vietnam (GU358515; GU358516; AB175082; AB175083; JX181935). According to the latest molecular research results, “*C. attenuata*” distributed in Vietnam should be an undescribed species [16]. Therefore, the *C. attenuata* species complex distributed in mainland China includes four reported species, namely, *C. attenuata*, *C. tanakae*, *C. anhuiensis* and *C. dongyangjiangensis*, as well as a cryptic species distributed in Guangxi.

In mainland China, *C. tanakae* has been confirmed to be more widely distributed than *C. attenuata*, which is only found in Western and Southeastern China [30]. However, the sample of *C. attenuata* in the abovementioned study included some samples of *C. anhuiensis*, because *C. anhuiensis* was considered a distinct species in subsequent studies [32]. Therefore, the distribution of *C. attenuata* needs to be re-evaluated in the literature. Only one report exists, to date, on the distribution of *C. anhuiensis*, and it is found only in Huangshan in the Anhui Province [32]. *C. dongyangjiangensis* is reported to be distributed in Chun’an County and Dongyang City in the Zhejiang Province [31]. In addition, a study on the synonyms of *C. dongyangjiangensis* exists, which also involves its distribution in Huangshan, Anhui Province [49].

In this study, we revised the distribution of four species in mainland China based on the most abundant sample of the *C. attenuata* species complex (Figure 1). *C. attenuata* is only distributed around the Sichuan Basin in China, especially in the west and northeast of the Sichuan Basin. *C. tanakae* is widely distributed in Southern mainland China. *C. anhuiensis* is distributed in Southeastern China, including Anhui, Zhejiang, Fujian, Jiangxi, Hunan and Guangdong Provinces. The distribution of *C. dongyangjiangensis* is basically sympatric with that of *C. anhuiensis*, including Anhui, Zhejiang, Jiangxi, Hunan and Guangdong Provinces.

### 4.2. The Confusion in the Taxonomy of the Crocidura attenuata Species Complex

In the present study, the revised distribution area of *C. attenuata* is limited to only around the Sichuan Basin. Therefore, the widely distributed “*C. attenuata*” mentioned in previous reports may be *C. anhuiensis*, *C. tanakae* or other cryptic species, and the results of the *C. attenuata* species complex in these reports also require further modifications. For example, (1) in our previous study conducted on the distribution and karyotype of *C. attenuata* [30,34], some samples of “*C. attenuata*” should have been *C. anhuiensis*, which have been corrected in this study (Appendix A). In addition, (2) regarding the morphological identification characteristics of *C. attenuata* [50] and the description of the morphological differences between *C. attenuata* and *C. tanakae* shrews [30,51], the samples of *C. attenuata* used were mixed with *C. anhuiensis* and the cryptic species in Vietnam, so these results also need to be verified further. Furthermore, (3) Zhang et al. [31] described a new species of *C. anhuiensis* obtained from China based on molecular and morphological data. In their report, the molecular specimen of “*C. attenuata*” was a cryptic species collected from Vietnam, and the molecular specimen of “*C. attenuata*” should have been *C. tanakae,* which is sympatrically distributed with *C. anhuiensis*. Although the taxonomic results of *C. anhuiensis* in this study are questionable, our results support the status of *C. anhuiensis* as a valid species. In addition, other studies have been conducted on the misuse of specimens involving the *C. attenuata* species complex [24,26,28,46,49], which have not been presented.

The two main reasons leading to the taxonomic confusion are as follows: first, most of the previous research only relied on morphology, which led to the misclassification of some species that were very morphologically similar. The second reason is that some small mammals lack extensive attention, leading to insufficient field investigations. Therefore, we appeal increasing the investigation and protection of these under-researched animals, and minimizing the harm to these animals during research [52].

### 4.3. The Taxonomic Effectiveness of Molecular, Karyotype and Morphological Methods

For the integrative taxonomic case of the *C. attenuata* species complex, only the molecular method can rapidly and effectively identify the species because all species can gather into an independent clade in the phylogenetic tree and obtain considerable support. The interspecific genetic distance also supports the valid species status of each species.

Although the karyotypic method is considered to be the most consistent research method for the concept of species and plays an important role in many taxonomic studies [7,8,53], it does not play a significant role in the taxonomy of the *C. attenuata* species complex and can only classify some specimens of *C. tanakae* according to its unique karyotype. There are two opposite extremes in the karyotype of the genus *Crocidura* in China. One is that the karyotype of the genus is conservative. This is to say that most species in the genus have the same karyotype (2 n = 40, FN = 54), and *C. attenuata*, *C. anhuiensis* and *C. dongyangjiangensis* all belong to this karyotype (Appendix A). However, the karyotypes in *C. tanakae* showed a high degree of polymorphism [34]. Therefore, it is inevitable to erroneously perform the species identification of the *C. attenuata* species complex based only on the karyotype method. For example, *C. tanakae* was promoted to a distinct species based on the karyotype difference between *C. attenuata* and *C. tanakae* [24], and this karyotype difference has been proven to be the result of a karyotype polymorphism occurring in *C. tanakae* [34]. In small mammals, karyotype polymorphisms may be a relatively common situation [54], which also poses a challenge to taxonomy based on karyotypes.

Morphological methods have always been considered the most intuitive and effective classical taxonomic methods, but some difficulties remain in the taxonomy of the *C. attenuata* species complex with a conservative morphology. In this study, we observed that with the increase in the sample size and distribution range of species, some interspecific morphological differences reported in previous studies have become intraspecific variations, especially the morphological differences reported among *C. attenuata*, *C. tanakae* and *C. anhuiensis* [30,31,52]. There are obvious molecular differences among species of the *C. attenuata* species complex, but there are no obvious interspecific morphological differences. We speculated that the reason for this was that *Crocidura* has experienced rapid evolution [16], which led to species differentiation without synchronous morphological variations [55].

## 5. Conclusions

In conclusion, there may be five species of the *C. attenuata* species complex distributed in mainland China, comprising of *C. attenuata*, *C. tanakae*, *C. anhuiensis, C. dongyangjiangensis* and a cryptic species distributed in Guangxi. The four reported species were revised and we observed that *C. attenuata* was distributed in the northeast and west around the Sichuan Basin in Western China, while *C. tanakae* was distributed throughout the Southern mainland of China. The distribution ranges of *C. anhuiensis* and *C. dongyangjiangensis* almost overlapped and were distributed in Southeast China. In the taxonomy of the sibling species with a conservative morphology, such as the *C. attenuata* species complex, molecular methods can often play an effective role and rapidly produce desirable results. Although the molecular classification threshold is not uniform, the integrated taxonomy combined with morphology, karyotype or other research methods will achieve more comprehensive and accurate results.

## Figures and Tables

**Figure 1 animals-13-00643-f001:**
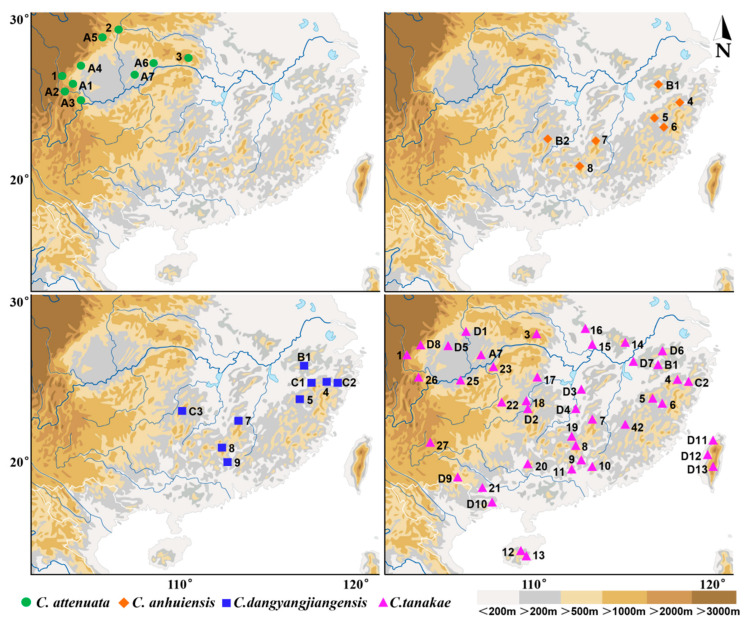
Specimen locality of *Crocidura attenuata* species complex in China.

**Figure 2 animals-13-00643-f002:**
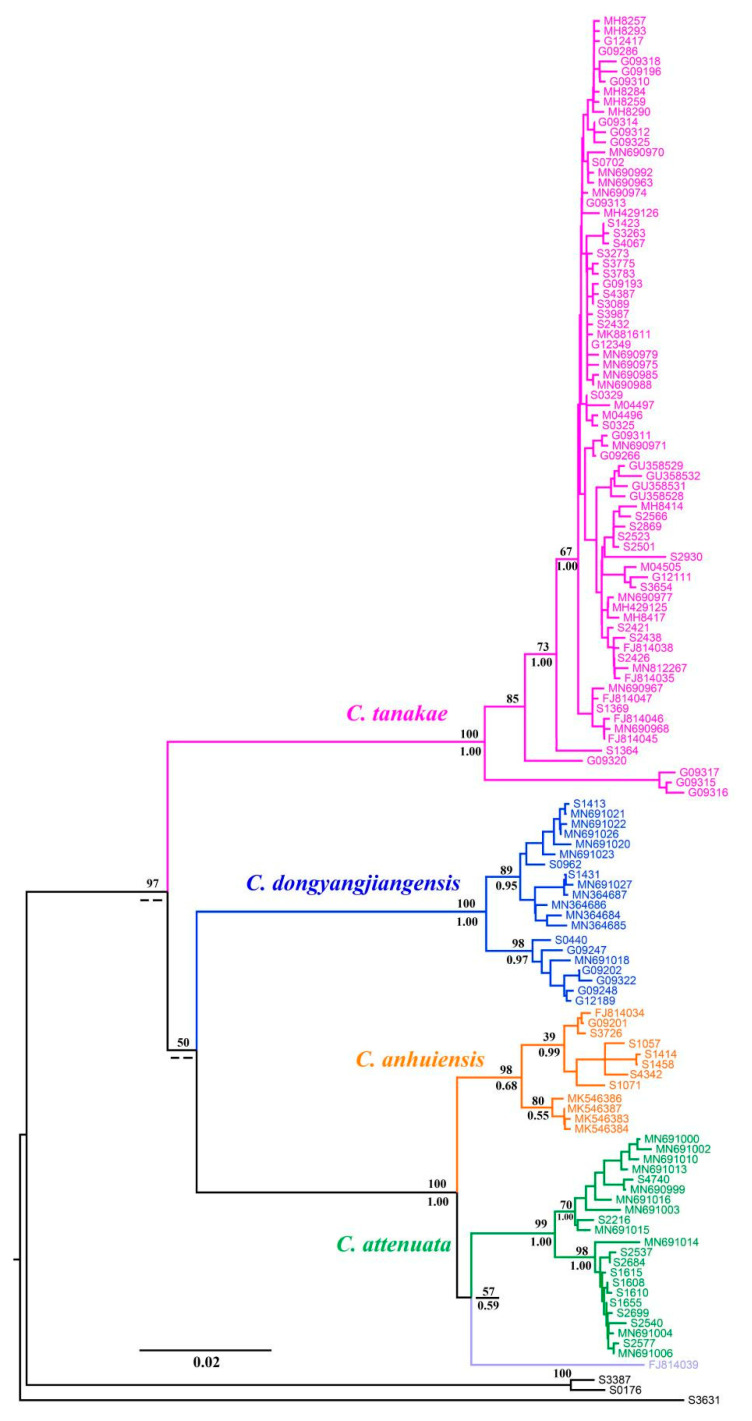
Phylogenetic tree of *C. attenuata* species complex based on *Cytb* gene. Numbers above and below branches represent bootstrap support (BS) of ML and posterior probabilities (PP) of BI, respectively.

**Figure 3 animals-13-00643-f003:**
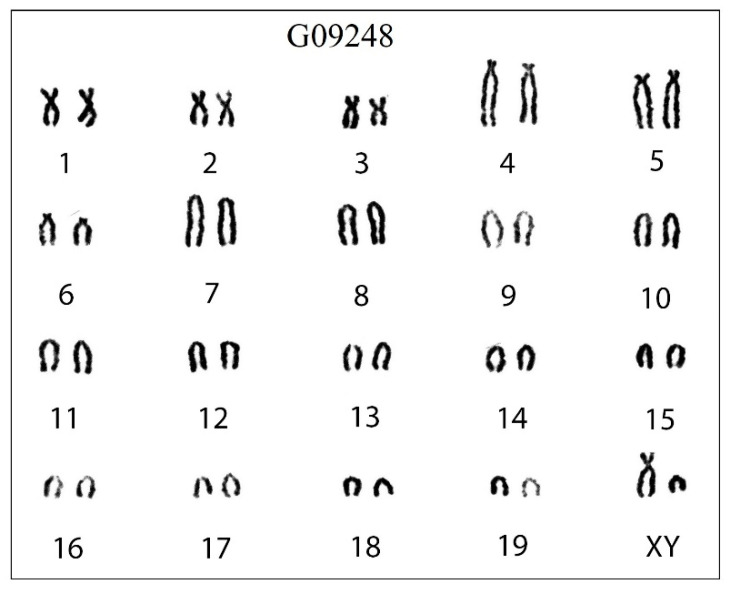
The karyotype of *C. dongyangjiangensis* in mainland China. The diploid chromosome number (2 n) is 40 and fundamental arm number (FN) is 54, with three pairs of metacentric or submetacentric (m/sm), three pairs of subtelocentric (st), and 13 pairs of acrocentric autosomes (t), a submetacentric X chromosome (X), and an acrocentric Y chromosome (Y). The order of the chromosomes is m/sm, st, t, X, Y and then from large to small chromosomes. The numbers 1–19 represent the number of chromosome pairs, and G09248 is the label of the specimen.

**Figure 4 animals-13-00643-f004:**
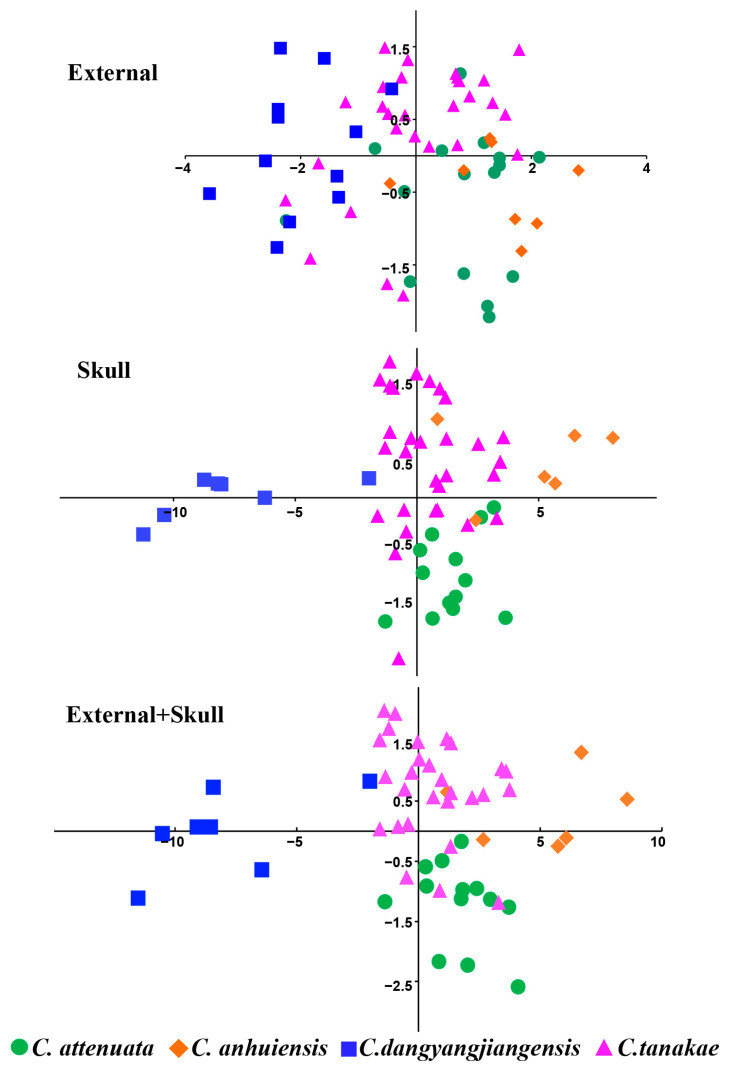
Principal component plots of *C. attenuata* species complex.

**Figure 5 animals-13-00643-f005:**
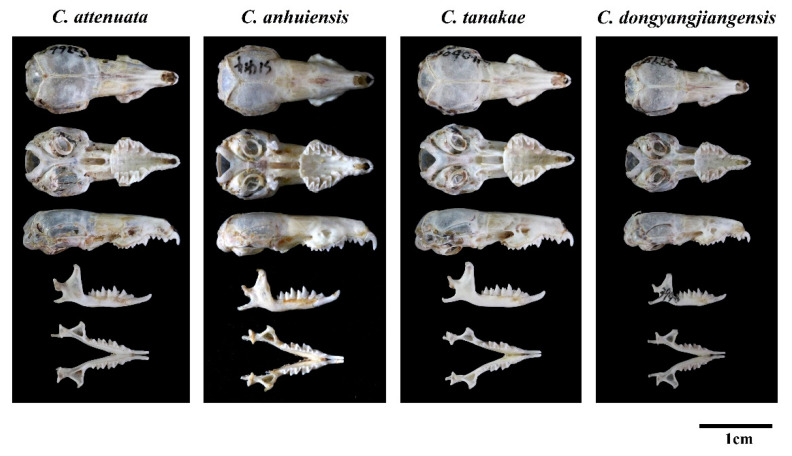
Skull photos of four species of *C. attenuata* species complex.

**Table 1 animals-13-00643-t001:** The interspecific and intraspecific genetic distances (*p*-distances) of *Crocidura attenuata* species complex based on mitochondrial cytochrome-*b* (*Cytb*) sequence data.

	*C. tanakae*	*C. dongyangjiangensis*	*C. anhuiensis*	*C. attenuata*
*C. tanakae*	0.0112 ± 0.0013			
*C. dongyangjiangensis*	0.128 ± 0.013	0.0149 ± 0.0023		
*C. anhuiensis*	0.129 ± 0.012	0.117 ± 0.012	0.0106 ± 0.002	
*C. attenuata*	0.136 ± 0.013	0.122 ± 0.012	0.041 ± 0.006	0.012 ± 0.0021

**Table 2 animals-13-00643-t002:** External and skull measurement indices of *C. attenuata* species complex.

Index	Item	*C. tanakae*	*C. dongyangjiangensis*	*C. anhuiensis*	*C. attenuata*
HB	Mean ± SD	76.31 ± 6.60	63.82 ± 6.54	75.75 ± 3.95	74.88 ± 6.54
	Range	64.00~95.00	55.00~78.50	69.00~82.50	61.00~85.00
	N	29	11	8	16
Tail	Mean ± SD	52.29 ± 5.66	48.32 ± 4.11	58.75 ± 5.87	56.56 ± 4.80
	Range	44.00~63.50	40.00~54.00	49.00~65.00	50.00~67.00
	N	28	11	8	16
Ear	Mean ± SD	8.45 ± 1.29	7.70 ± 1.22	8.65 ± 0.68	7.96 ± 1.08
	Range	5.76~11.06	5.58~9.44	7.93~9.85	6.05~9.68
	N	29	11	8	16
HF	Mean ± SD	12.98 ± 0.66	11.39 ± 0.50	14.77 ± 0.71	14.19 ± 1.10
	Range	12.01~14.44	10.35~11.96	13.84~15.66	11.74~16.53
	N	30	11	8	16
CIL	Mean ± SD	20.20 ± 0.64	17.37 ± 0.88	21.34 ± 0.77	21.06 ± 0.56
	Range	19.28~21.63	16.53~19.14	20.43~22.44	20.04~22.10
	N	30	7	7	14
MTR	Mean ± SD	5.82 ± 0.15	4.89 ± 0.32	6.25 ± 0.22	5.88 ± 0.12
	Range	5.59~6.13	4.54~5.58	5.91~6.51	5.67~6.07
	N	30	11	8	16
HCC	Mean ± SD	4.96 ± 0.20	4.43 ± 0.30	5.15 ± 0.16	5.03 ± 0.14
	Range	4.29~5.36	3.98~4.89	4.97~5.40	4.83~5.33
	N	30	8	8	15
RW	Mean ± SD	2.48 ± 0.14	2.02 ± 0.15	2.62 ± 0.15	2.30 ± 0.16
	Range	2.17~2.72	1.82~2.36	2.40~2.86	2.07~2.66
	N	30	11	8	16
MB	Mean ± SD	6.64 ± 0.47	5.46 ± 0.33	7.02 ± 0.35	6.47 ± 0.28
	Range	5.95~8.53	5.07~6.24	6.46~7.37	5.67~6.89
	N	30	11	8	16
IO	Mean ± SD	4.43 ± 0.16	3.93 ± 0.20	4.90 ± 0.24	4.55 ± 0.15
	Range	4.07~4.80	3.66~4.29	4.51~5.20	4.24~4.80
	N	30	10	8	15
GW	Mean ± SD	9.13 ± 0.30	8.04 ± 0.35	9.80 ± 0.40	9.45 ± 0.29
	Range	8.62~9.74	7.41~8.50	9.28~10.41	8.97~10.09
	N	30	9	7	15
PIL	Mean ± SD	9.28 ± 0.30	7.71 ± 0.58	9.89 ± 0.50	9.56 ± 0.29
	Range	8.69~9.91	6.78~8.99	9.24~10.68	9.01~10.18
	N	30	10	7	16
PAL	Mean ± SD	7.83 ± 0.25	6.56 ± 0.48	8.36 ± 0.37	7.99 ± 0.27
	Range	7.36~8.36	6.08~7.66	7.84~8.98	7.41~8.44
	N	30	9	8	16
PPL	Mean ± SD	8.94 ± 0.40	7.72 ± 0.22	9.50 ± 0.37	9.45 ± 0.31
	Range	8.16~9.70	7.42~8.14	8.90~10.05	8.88~10.02
	N	30	7	8	13
UTR	Mean ± SD	8.93 ± 0.29	7.61 ± 0.46	9.61 ± 0.41	9.21 ± 0.24
	Range	8.44~9.58	7.06~8.51	9.02~10.13	8.75~9.58
	N	30	11	7	16
P^4^-M^3^	Mean ± SD	5.12 ± 0.15	4.40 ± 0.28	5.57 ± 0.22	5.16 ± 0.12
	Range	4.88~5.42	4.04~4.98	5.26~5.88	5.00~5.37
	N	30	11	8	16
PW1	Mean ± SD	6.17 ± 0.25	5.23 ± 0.33	6.59 ± 0.28	6.19 ± 0.20
	Range	5.76~6.70	4.82~6.03	6.03~6.86	5.86~6.65
	N	30	10	8	15
PGL	Mean ± SD	6.42 ± 0.23	5.57 ± 0.35	6.80 ± 0.34	6.49 ± 0.16
	Range	6.03~7.04	5.21~6.39	6.32~7.22	6.23~6.87
	N	30	10	8	15
LDT2	Mean ± SD	8.18 ± 0.27	6.92 ± 0.43	8.86 ± 0.41	8.47 ± 0.19
	Range	7.74~8.75	6.38~7.90	8.19~9.36	8.22~8.89
	N	30	11	8	16
LDT1	Mean ± SD	6.06 ± 0.22	5.24 ± 0.31	6.53 ± 0.44	6.20 ± 0.14
	Range	5.59~6.47	4.93~5.92	5.51~6.87	6.02~6.51
	N	30	11	8	16
M_1_-M_3_	Mean ± SD	4.19 ± 0.14	3.65 ± 0.22	4.56 ± 0.18	4.24 ± 0.11
	Range	3.91~4.44	3.37~4.11	4.23~4.74	4.10~4.43
	N	30	11	8	16
BCP	Mean ± SD	1.07 ± 010	0.86 ± 0.09	1.16 ± 0.10	1.01 ± 0.10
	Range	0.86~1.30	0.72~0.95	1.01~1.32	0.83~1.21
	N	30	11	8	16
ML	Mean ± SD	12.68 ± 0.43	10.70 ± 0.64	13.78 ± 0.70	13.23 ± 0.41
	Range	12.06~13.71	9.86~12.10	12.79~14.81	12.37~13.92
	N	30	10	8	16
COR	Mean ± SD	4.84 ± 0.23	4.06 ± 0.26	5.29 ± 0.30	4.90 ± 0.20
	Range	4.38~5.33	3.57~4.47	4.81~5.81	4.58~5.33
	N	30	11	8	16

Note: HB: head and body length, Tail: tail length, Ear: ear length, HF: hind foot length, CIL: condylo-incisive length, HCC: height of cranial capsule, RW: rostrum width, MB: maxillary breadth, IO: least interorbital width, GW: greatest width of skull, UTR: upper toothrow length, P4–M3: length of anterior tip of P4 to posterior border of M3, b PW1: breadth of palate between the buccal margins of second molars, PGL: postglenoid width, M1–M3:length of lower molar series, ML: length of mandible from tip of incisor to posterior edge of condyle, COR: height of coronoid process, MTR: length of maxillary tooth row, PAL: palatilar length, PPL: post-palatal length, LDT1: length of dentary teeth excluding incisors, LDT2: length of dentary teeth including incisors, PIL: palato-incisor length and BCP: breadth of coronoid process. N is the abbreviation of specimen number.

## Data Availability

All the sequences used in this study were accessed through the GenBank database and the accession numbers are listed in Appendix A. Morphological specimens and chromosome materials were deposited in the Zoological Research Team at Marine College, Shandong University (Weihai).

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
