# Peer review of "The Effectiveness of Molecular, Karyotype and Morphological Methods in the Identification of Morphologically Conservative Sibling Species: An Integrative Taxonomic Case of the Crocidura attenuata Species Complex in Mainland China"

_animals, 2023, doi:10.3390/ani13040643_

Round 1
Reviewer 1 Report
This article integrated molecular, karyotype and morphological data to classify the species complex of grey musk shrew, revealing the convenience of molecular methods and the advantages of integrated taxonomy in the closely related species with conservative morphology. It is a research with detailed data, reliable results and guiding significance for species complex classification. Please note the following points:
1. Introduction is missing some relevant references. Line 58-60…
2. please provide a clear figure for publication. Figure 1, 4.
3. The univariate analysis of variance and interspecific comparison in morphological analysis are not described in the method;
4. Is the ethics certificate numbered? If yes, I hope it can be reflected in the text.
5. There are some small problems: N in Table 2 is the abbreviation of number, which should be explained in the table footer; the citation “Maddalena and Ruedi 1994 ”mentioned in Table S3 are not included in the references.
6. the manuscript needs extensive revision by native speaker.
Author Response
We thank you very much for your highly professional advices and questions, which is very helpful to the improvement of this manuscript.
- Introduction is missing some relevant references. Line 58-60…
>>> Thank you for your suggestion. There are many relevant references about the application of integrated taxonomy in species complex, so we have listed four representative references.
- please provide a clear figure for publication. Figure 1, 4.
>>> According to your advice, we have replaced more high-definition Figure 1, 4.
- The univariate analysis of variance and interspecific comparison in morphological analysis are not described in the method;
>>> According to your advice, We have added descriptions of these analyses to the method.
- Is the ethics certificate numbered? If yes, I hope it can be reflected in the text.
>>> We provide the ethics certificate number in the article.
- There are some small problems: N in Table 2 is the abbreviation of number, which should be explained in the table footer; the citation “Maddalena and Ruedi 1994 ”mentioned in Table S3 are not included in the references.
>>> According to your advice, we modified the relevant context.
- the manuscript needs extensive revision by native speaker.
>>> According to your advice, we have revised the language through MDPI English Editing, the English editing ID is English-59060.
Reviewer 2 Report
In this article, the authors use molecular, karyotype and morphological methods to revise the classification and distribution of the C. attenuata species complex and discussed the effectiveness of the integrative methods. They did a lot field works spanning about 20 years and used enough specimens. Overall, the study is well conducted. The language is adequate and easily understood.
- Several suggestions I would like to put forward:
1. Line 59-60: When citing the mammal diversity database, it will be better to attach the site URL.
2. Line 99-102: avoid using too long sentences.
3. Line 204: complexbased --- complex based
4. Line 205: The numbers are not only “above” branches, please reconsider it.
5. Line 323-326: I think the author intend to express the similarity between C. attenuata and C. anhuiensis in morphology, karyotype, and genetic distance. So, the word “although” may be deleted. Also, because the word “However” is in the beginning of the next sentence.
Besides, some other minor mistakes in the article are not mentioned here, thus, the authors should read the paper thoroughly and check it carefully.
Author Response
We thank you very much for your highly professional advices and questions, which is very helpful to the improvement of this manuscript.
- Several suggestions I would like to put forward:
1. Line 59-60: When citing the mammal diversity database, it will be better to attach the site URL.
>>> According to your advice, we added the website URL (https://www.mammaldiversity.org/).
- Line 99-102: avoid using too long sentences.
>>> According to your advice, we have split the long sentence.
In this study, molecular, karyotype and morphological methods were integrated to conduct taxonomic research on the C. attenuata species complex to revise its species classification and distribution. Moreover, the effects of different research methods on the taxonomy of related species with a conservative morphology were discussed.
- Line 204: complexbased --- complex based
>>> According to your advice, we modified the relevant context.
- Line 205:The numbers are not only “above” branches, please reconsider it.
>>> According to your advice, we modified the relevant context.
Numbers above and below branches represent bootstrap support (BS) of ML and posterior probabilities (PP) of BI respectively.
- Line 323-326: I think the author intend to express the similarity between C. attenuata and C. anhuiensis in morphology, karyotype, and genetic distance. So, the word “although” may be deleted. Also, because the word “However” is in the beginning of the next sentence.
>>> According to your advice, we modified the relevant context.
Besides, some other minor mistakes in the article are not mentioned here, thus, the authors should read the paper thoroughly and check it carefully.
>>>Thank you for your suggestion. We must carefully review and revise the manuscript.
Reviewer 3 Report
The MS deals with the characterization of three species of white toothed shrew Crocidura spp. Given that little is known on these species and given the large sample size analysed in this work, the MS deserves to be published. I only ask for some minor revisions:
1. Why were Crocidura specimens killed? Please, take a look at this paper, which may provide different positions on this topic: Russo D., Ancillotto L., Hughes A.C., Galimberti A., Mori E. (2017). Collection of voucher specimens for bat research: conservation, ethical implications, reduction, and alternatives. Mammal Review, 47: 237-246.
2. Lines 117-138. How long was the amplified fragment?
3. Figure 1 is unreadable, please improve its quality.
4. Discussion should include some part on the conservation of these poorly studied species.
Author Response
We thank you very much for your highly professional advices and questions, which is very helpful to the improvement of this manuscript.
- Why were Crocidura specimens killed? Please, take a look at this paper, which may provide different positions on this topic: Russo D., Ancillotto L., Hughes A.C., Galimberti A., Mori E. (2017). Collection of voucher specimens for bat research: conservation, ethical implications, reduction, and alternatives. Mammal Review, 47: 237-246.
>>> Thank you for your suggestion. We added the advice to reduce the harm to animals in the discussion, and we will try to reduce the harm to animals in future research. However, this study involved morphological and karyotype methods, so some animals had to be killed. At the same time, we strictly follow the animal welfare policy when handling specimens.
- Lines 117-138. How long was the amplified fragment?
>>> We have amplified the full length of Cytb gene, a total of 1140bp, and added this content to the text.
- Figure 1 is unreadable, please improve its quality.
>>> According to your advice, we have replaced more high-definition Figure 1.
- Discussion should include some part on the conservation of these poorly studied species.
>>> According to your advice, We added this content in the discussion, that is, we call for increasing attention and protection to these animals.